# Disulfiram (Tetraethylthiuram Disulfide) in the Treatment of Lyme Disease and Babesiosis: Report of Experience in Three Cases

**DOI:** 10.3390/antibiotics8020072

**Published:** 2019-05-30

**Authors:** Kenneth B. Liegner

**Affiliations:** 1592 Route 22—Suite 1B, Pawling, NY 12564, USA; 2Northwell System, Northern Westchester Hospital, Mount Kisco, NY 10549, USA; 3Health Quest System, Sharon Hospital, Sharon, CT 06069, USA; kbliegnermd@optonline.net; Tel.: +1-845-493-0274; Fax: +1-845-493-0279

**Keywords:** Chronic Lyme disease, neuroborreliosis, Post-treatment Lyme disease syndrome (PTLDS), relapsing babesiosis, persistent infection, antibiotic treatment failure, disulfiram

## Abstract

Three patients, each of whom had required intensive open-ended antimicrobial therapy for control of the symptoms of chronic relapsing neurological Lyme disease and relapsing babesiosis, were able to discontinue treatment and remain clinically well for periods of observation of 6–23 months following the completion of a finite course of treatment solely with disulfiram. One patient relapsed at six months and is being re-treated with disulfiram.

## 1. Introduction

Relapsing chronic and neurological Lyme disease with persistent infection despite treatment is well-documented and problematic [1,2,3,4,5,6,7,8,9,10,11,12,13,14,15,16]. Mechanisms for this remarkable phenomenon have been proposed, including intra-cellularity, complexity of life cycle, biofilm formation, privileged niches, and host immune suppression [17,18,19,20,21,22,23,24,25,26,27,28,29] (Appendix A). It has become increasingly clear that treatment of a borrelial infection with the currently available antimicrobial agents may suppress but not eradicate the infection. Syphilis provides a precedent for this [30,31]. Babesiosis also has the potential to persist despite the best available treatment methods, particularly in immune-compromised hosts [32,33,34]. It has been suggested that co-infection with Lyme disease and babesiosis may result in more severe disease expression [35]. Several groups of researchers have recently identified novel compounds for Lyme disease treatment [26,36,37]. Disulfiram was identified as a highly active compound against borrelia *in vitro*, after screening 7450 drug molecules from different chemical libraries [36,38]. This finding suggests that United States Food & Drug Administration FDA-approved disulfiram might have the potential to treat Lyme disease in humans. Reported here are three patients with relapsing borreliosis, each of whom happened to also have babesiosis. The application of disulfiram eliminated the need for further antimicrobial treatment during the period of observation in two of the three patients, although one patient relapsed and is being re-treated. Disulfiram, unexpectedly, appeared to exert activity against not only borreliosis but also babesiosis.

## 2. Case 1

In late May 2008, shortly after having been on an overgrown bike trail in Setauket in Suffolk County, New York, and having sat on a nearby pile of logs, the patient noted a tiny black ‘speck’ on his inner right thigh. A few days later he noticed it again and scratched it, whereupon it burst, with a release of a small amount of blood. In mid-July 2008 he began to feel sick, with a fever of 102 degrees Fahrenheit and light sensitivity. He consulted his primary care practice, where he was seen by a physician’s assistant who noticed an inflamed 3-inch diameter bull’s eye rash on his upper inner right arm—where the patient had previously noted a coin-sized area of redness—and diagnosed the patient with Lyme disease. He was given a prescription for amoxicillin for three weeks. The evening of the first day of treatment, a severe headache developed with chills, drenching sweats, and fever to 103 degrees Fahrenheit. One week later, while still on amoxicillin, he felt he was not getting well, with an ongoing headache and ‘cloudy-headedness’. Treatment was changed from amoxicillin to doxycycline (100 mg, by mouth (*per os* or PO) every [*Quaque* or Q] 12 h). Two weeks later, he returned to the physician’s assistant with continued headache and was referred to an infectious diseases physician and a neurologist. A spinal fluid examination in August 2008 revealed no pleocytosis but a slightly elevated cerebrospinal fluid CSF protein. The Lyme polymerase chain reaction PCR in CSF was negative. The Serum Lyme immunoglobulin M IgM was positive at Immugen Laboratories, Norwood, Massachusetts IMUGEN (14.4 (nl < 1)) with negative immunoglobulin G and immunoglobulin A IgG and IgA. CSF IgM at IMUGEN was elevated at 1.4 (nl < 1). The Lyme IgM Western blot was positive. Tests for babesiosis, *Ehrlichia chaffeensis*, *Anaplasma phagocytophilum*, and eastern equine encephalitis were negative. The electroencephalogram and brain magnetic resonance imaging (MRI) were normal. Tinnitus, ‘electric shock’-like sensations, muscle fasciculations, and migratory joint pain developed. 

In October 2008 he was hospitalized and underwent a second lumbar puncture. Again, there was no pleocytosis. Serum Lyme IgM was positive again, at IMUGEN (IgM > 9.4 (nl < 1)), with negative CSF IgM. CSF protein again was modestly elevated (51 mg%–upper limit of normal ULN 45 mg%). He received five days of intravenous (IV) ceftriaxone in hospital and an additional 23 days as an out-patient. Treatment improved, but did not fully resolve, the symptoms.

In November 2008, a physician experienced with tick-borne infections was consulted. Further oral antibiotics were prescribed, but despite these his condition worsened. Intravenous ceftriaxone, 2 g/day, was reinstituted from June to October of 2009. Joints were better but not all symptoms resolved, and the patient felt he had ‘plateaued’ but had not got completely well. He felt a sense of pressure within his skull and sensations he described “as though worms were crawling through my face and gold fish were swimming through my brain”. 

He was treated empirically for the possibility of co-infection with babesiosis, with azithromycin and atovaquone for five months [39]. Following this treatment, his chills resolved.

Levaquin was prescribed at the patient’s request for several months, but without impact or benefit.

He returned to his primary care physician in 2010. PenVK (500 mg PO] Q 8 h) was initially prescribed, but pain returned in his shoulders, hips, and other joints, and the dosage was gradually increased to a maximum of 8 g PO three times daily TID, with 500 mg probenicid PO TID, in order to control his symptoms. Although these doses helped, he was still experiencing joint pain. At the patient’s request, treatment was changed from high dose PenVK to minocycline, with the dose increased to as high as 300 mg PO twice daily BID, which improved his symptoms [40]. The patient added banderol and samento, with subjective benefit [41]. Joint pain lessened except in his feet. His thinking was improved, and tinnitus became only intermittent. 

When first seen June 2011, this then 41-year-old engineer was complaining of headaches, joint pain, involving shoulders, wrists, fingers, hips, and ankles, and head pressure. Although he felt that he was doing better cognitively compared to his prior status, he still experienced some difficulty thinking and reading. Pan-hypogammaglobulinemia was noted (IgG 452 mg% (lower limit of normal LLN 694); IgA 51 (LLN 81); IgM 20 (LLN 48)). IgG sub-classes 1–3 were reduced. Surprisingly, babesia fluorescent-in-situ-hybridization (F.I.S.H.) returned positive for detection of the ribonucleic acid (RNA) of babesia piroplasms within the red blood cells at IGeneX Laboratory, Milpitas, California (IGeneX), despite the patient having previously received five months of empirical treatment with azithromycin and atovaquone aimed at a possible babesia co-infection. Hepatic transaminases were mildly elevated.

Minocycline was discontinued and treatment with amoxicillin (8750 mg PO Q 12 h) was initiated. The peak amoxicillin level in August 2011 was 19.5 mcg/mL (goal for central nervous system CNS treatment = 15–25 mcg/mL) [42,43,44] (Appendix A). Azithromycin (1200 mg PO QD) and atovaquone (750 mg PO BID) were added to his regimen, in view of the positive babesia F.I.S.H. Tinidazole (500 mg PO BID) limited to two consecutive days per week only, was added in April 2012, aimed at cystic phase borreliae [45].

An immunologist was consulted September 2011, who confirmed pan-hypogammaglobulinema and an inadequate response to the *S. pneumonia* vaccination and treated the patient for five months with IV immunoglobulin therapy, ending in summer 2012. Although the patient stated he did not experience benefits from this therapy, the numbness in his feet resolved on this, coincident with intensive oral antibiotic and anti-piroplasm treatment. The role of immune-impairment in his subsequent chronic course of illness needs to be considered.

By July 2012, the patient reported feeling overall improved and, although not completely well, was relatively satisfied with his status, with minimal joint pain and paresthesia and improved cognitive functioning. He still occasionally experienced bizarre sensations of worm-like movement in his face without visible surface fasciculations.

October 2012, he began to experience increased joint pain and pinprick-like sensations. He reported that he was under considerable job and personal stress at the time, and wondered if this could be responsible for his recently increased symptoms. In any case, he found that when he increased his dose of amoxicillin on his own advice, these subsided. He was given latitude to increase his amoxicillin dose to 12,250 mg PO Q 12 h and did well on this dose, at which the peak amoxicillin level, subsequently measured March 2013, was 28 mcg/mL. Azithromycin and atovaquone were discontinued and clarithromycin (1500 mg PO BID), hydroxychloroquine (200 mg PO BID) [46], atovaquone (250 mg) and proguanil (100 mg X 2 PO BID) [47], and amitriptyline (20 mg at bed time) were prescribed. Tinidazole was discontinued and metronidazole [48] (500 mg PO BID, for two consecutive days/week) was substituted, the latter change necessitated by insurance reimbursement constraints. 

In November 2012 the patient reported feeling greatly improved, commenting, “something is going right!”

The patient remained relatively well on a regimen of amoxicillin, clarithromycin, hydroxychloroquine, metronidazole, atovaquone/proguanil, and amitriptyline, until August 2013, when he decided to discontinue clarithromycin, hydroxychloroquine, and amitriptyline.

He found that sleep deprivation and poor diet exacerbated his symptoms and took measures to rectify both, with improved status.

In October 2013, he experienced a significant recurrence of musculoskeletal pain. He commented that with past attempts to decrease or discontinue antimicrobial treatment, his symptoms “came roaring back like a freight train”, but he preferred not to resume hydroxychloroquine, clarithromycin, or amitriptyline.

Instead, the decision was made to resume treatment with minocycline, but limited to 200 mg PO BID, and to continue high dose amoxicillin, atovaquone/proguanil, and metronidazole (the latter for two days/week only) until December 2016. Ample probiotics were used, and surveillance labs remained satisfactory throughout. Overall, he found this regimen to be satisfactory in controlling symptoms, many of which, nonetheless, were still present, but at a reduced intensity. He was able to function at work and at home and was able to resume exercise, including some weight-lifting. He expressed feeling generally ‘decent’ and estimated that on this intensive regimen of oral treatment, he was at some 90% of his pre-morbid state of wellness. He commented, though, in May 2015, that had he not been able to be treated intensively during the early course of his illness when he was overwhelmed by the onslaught of an intense and evolving slew of multi-system symptoms, that he might well have committed suicide.

In December 2016, the patient decided to try to adopt a ‘pulse’ regimen with two–three weeks on treatment and two–three weeks off treatment, with a view to trying to taper off and discontinue treatment all together, if possible. He discovered that his symptoms recrudesced within two weeks of cessation of treatment, and he concluded that he would be unable to successfully discontinue the regimen of treatment. He commented that he ‘felt like jumping off a bridge’, exasperated by struggling with his illness for almost 10 years, with no end in sight.

In March 2017, the patient reported having viewed a recording of a presentation on Lyme disease by a microbiologist at an academic conference, during which it was stated that, *in vitro*, disulfiram was the most highly active agent against *B. burgdorferi*, bar none [49]. The speaker cited previously published data by *Pothineni* et al., which showed that disulfiram can eliminate *B. burgdorferi* very effectively at a minimum inhibitory concentration MIC of 0.625 micromoles (μM) [36]. 

The patient requested a trial of treatment with disulfiram. After a full discussion, which included the risks of disulfiram use and its uncertain benefits in the treatment of Lyme disease in patients, amoxicillin, minocycline, atovaquone/proguanil, and metronidazole were discontinued, and a 90-day course of disulfiram (500 mg/day) was started, in March 2017. Periodic surveillance labs were obtained and were satisfactory throughout. The patient reported on his status by letter every few weeks and tolerated treatment well. 

The patient left a telephone message, 13 July 2017, cancelling his scheduled follow-up and declaring, “I’m cured”. He mentioned in passing that he had required hospitalization. Upon subsequent telephone query 19 July 2017, the patient stated he was well and clarified that the hospitalization had been required for psychiatric reasons. 

It was pointed out to the patient that disulfiram has occasionally been reported to cause psychiatric disturbance [50,51,52]. The patient was dubious that disulfiram had a role in his need for psychiatric hospitalization, as he had been under very significant situational stress. However, he averred, “even if it *was* responsible it was *worth* it! Don’t fail to offer it to people just because of *that*!”

The patient has been contacted by telephone for a follow-up and has remained apparently clinically well both medically and psychiatrically, with no antimicrobial or psychopharmacotherapy needed for 23 months. 

## 3. Case 2

First seen in July 2010, this then 74year-old retired physician, a resident of eastern Pennsylvania, had a history of diabetes, coronary artery disease, depression, and lumbar spinal stenosis. He lived on 20 acres of woods and fields, had four indoor/outdoor dogs, and reported some 100 known tick attachments during the course of his life, but no history of any rash suspicious for erythema migrans. In late May 2009, he had an acute febrile illness, with a temperature reaching 102 degrees Fahrenheit, for which he was admitted to hospital and given a short course of an intravenous antibiotic. The fever resolved, and he was discharged. No etiologic diagnosis was established at that time, and a test for Lyme disease was negative. In August 2009, a repeat test for Lyme disease was strongly positive, and he was treated with doxycyline for one month with noticeable improvement, although he was still unwell. He exhibited cognitive, gait, and balance problems, and was referred for infectious diseases and neurology consultations. He was found to have proximal muscle weakness, cerebellar ataxia, and thoracolumbar radiculopathy. His cerebrospinal fluid tested strongly positive for Lyme disease, with significantly elevated CSF protein but no pleocytosis. An MRI of the brain in October 2009 showed periventricular white matter disease, and what was termed ‘age appropriate’ parenchymal volume loss. A positron emission tomography/computer tomography PET/CT at that time showed diminished activity symmetrically in bilateral anteromedial temporal lobes and, to a lesser degree, posterior bilateral parietal lobes, and mild cerebral atrophy with differential diagnoses, including ‘senescent change, microvascular disease, and Alzheimer’s dementia’. A Lyme Western blot in serum showed 10 out of 10 IgG bands as well as a positive IgM blot. Thirty days of IV ceftriaxone was given, ending November 2009, with significant relief of symptoms, which relapsed after some five months. A CSF examination in August 2010 at the Laboratory for the Diagnosis of Lyme disease at the State University of New York at Stony Brook showed CSF Enzyme-linked immunosorbent assay ELISA to be positive, at 0.826, with a serum result of 0.389, with a positive cut off of 0.155 and a CSF/serum ratio of 2.12. The IgM Lyme Western blot was positive, and the IgG blot showed 4 out of 10 United States Centers for Disease Control CDC-specific bands. The Lyme PCR was negative in the cerebrospinal fluid. In September 2010, DNA sequences specific for the detection of the outer surface protein A (OspA) plasmid of *Borrelia burgdorferi* were detected by PCR in the serum at IGeneX, confirmed by a dot blot assay. 

An additional 30-day course of IV ceftriaxone (2 g/day) was administered from September to October 2010, with improvement in multi-system symptoms, followed by treatment with oral minocycline (200 mg/day). He reported improved sleep, memory, ability to organize, and proximal muscle weakness notably improved. Despite the continuation of minocycline, by May 2011 his fatigue required afternoon naps and diminished stamina, and he curtailed driving his car or traveling.

He was observed off antibiotic therapy. An initial improvement in fatigue was followed by worsened muscle aches, hip discomfort, and a ‘dazed’ mental state. He remained in bed most of the time.

A lumbar puncture in July 2011 showed a CSF/serum index of 4.03, with positive IgG and IgM Western blots in CSF at the State University of New York at Stony Brook. 

A positive test for *Babesia microti* IgG was detected in July 2011, and a one-month course of atovaquone 250 mg/proguanil 100 mg was given as monotherapy, with which the patient reported feeling somewhat better.

Antibiotic therapy aimed at Lyme disease was withheld from mid-June to mid-September 2011. Proximal muscle weakness recurred in all four limbs. A repeat lumbar puncture in September 2011 showed six CDC-specific IgG bands in CSF with an index of 4.16, mildly elevated CSF proteins, and negative Lyme PCR in CSF.

By this time, the patient had received two prior 30-day courses of IV ceftriaxone, each time with benefits which, however, did not endure, and the patient clinically deteriorated despite extended oral antibiotic therapy with minocycline.

A fully implanted vascular access device (PORT) was placed, a short tapering course of prednisone was given for the myopathy, and IV ceftriaxone was re-instituted in late September 2011 and continued for 10.5 months, until August 2012, during which time the patient evidenced progressive improvement in energy, cognition, mood, physical strength, and stamina. Proximal myopathy and the broad-based gait that it had caused resolved within several months of the reinstitution of IV ceftriaxone. 

Tinidazole 500 mg twice daily for two consecutive days per week was added to the daily IV ceftriaxone at the 10th month [45]. A spinal fluid examination in July 2012 revealed a CSF/serum index of 7.81. The patient was able to resume full days of physical and mental activity and was feeling that he had achieved a ‘bench mark’ of near normal status for age by August 2012. 

Considering how he had been unable to remain well despite extended oral antibiotic therapy alone, the decision was made jointly with the patient to institute a rotating cycle of treatment consisting of 30 days of IV ceftriaxone (2 g/day), followed by 30 days of tinidazole (500 mg twice daily for two consecutive days/week), followed by 30 days of atovaquone (250 mg) and Proguanil (100 mg daily)—repeating this regimen as a 90-day cycle. On this regimen he was doing well and continued to gain ground.

In December 2012, the patient experienced malaise and shaking chills, and blood cultures were drawn from his PORT. These returned positive for gram negative rods and the patient was summoned to the emergency room of his local hospital, where he was found to be hypotensive. He was admitted to the intensive care unit, where he required fluid resuscitation, pressor support, and intravenous ciprofloxacin. An enterobacter species isolated from a peripheral blood culture was identical to that previously grown from the patient’s PORT, which was removed. The culture of the catheter tip grew out the same organism. He was discharged after five days and completed a course of oral ciprofloxacin.

Further antimicrobial treatment aimed at tick-borne illness was deferred and the patient observed.

Swelling and effusion of the right knee developed in January 2013.

The CSF/serum index in January 2013 was 7.44.

Until July 2013 the patient remained reasonably well, although not as well as he had felt prior to his last course of intravenous ceftriaxone, ending October 2012.

In July 2013, a trial of high dose oral amoxicillin (5.25 g Q 12 h) combined with tinidazole (500 mg PO twice daily for two consecutive days per week) was given for 30 days, with unclear impact.

In December 2013, the OspA plasmid target of *Borrelia burdgorferi* was detected by PCR in the serum once again, confirmed by a Southern dot blot at IGeneX.

He remained fairly well and stable until April 2014, when severe, debilitating fatigue and other symptoms, reminiscent to prior relapses of Lyme disease, recurred. Empiric retreatment with atovaqoune 250 mg/proguanil 100 mg daily notably improved his condition, and a cycle of rotating intramuscular ceftriaxone (1 g daily for 30 days), followed by 30 days of atovaquone, (250 mg) and proguanil (100 mg daily), followed by tinidazone (500 mg twice daily, for two consecutive days per week, for 30 days) for 90-day repeating cycles was instituted. He did well until July 2015 when, despite this regimen, he experienced a major depressive episode. The OspA plasmid target of *Borrelia burgdorferi* DNA was detected by PCR and confirmed by a southern dot blot at IGeneX yet again during this clinical relapse.

Psychopharmacotherapy was adjusted and intramuscular ceftriaxone was increased from 1 g to 2 g daily for 30 days of his 90-day cycle of 30 days sequences of ceftriaxone/atovaquone/tinidazole.

In January 2016 he was still experiencing significant fatigue and depression. Aripiprazole was added to his psychopharmacotherapy, which did increase his energy. His antimicrobial regimen was changed from sequential 30-day rounds of ceftriaxone, atovaquone/proguanil, and tinidazole treatment to a regimen of ongoing intramuscular ceftriaxone (2 g per day for five to six consecutive days per week, followed by two to three days off) with daily atovaquone (250 mg) and proguanil (100 mg), and tinidazole for two consecutive days per week, throughout.

His psychopharmacotherapy during this time this time included duloxetine, amphetamine, and aripiprazole. He was on low dose imipramine for treatment of bladder spasticity. Intravenous ketamine infusions every six weeks was added to the regimen in April 2016 for severe depression.

On this combined regimen of antimicrobial therapy and psychopharmacotherapy until December 2017, the patient stabilized, his psychiatric and physical symptoms progressively improved, and he was able to enjoy a reasonably good quality of life.

In July 2017, he was apprised of the experience of the patient described in Case 1. Over a period of several months, the patient pondered it carefully himself and discussed it with his physicians, and the physicians involved in his care held consultations between themselves. In view of the problematic nature of his illness and its treatment, and with discussion of the risks of disulfiram treatment and the extremely limited data on its utility in the treatment of patients with Lyme disease, a trial of treatment with disulfiram was undertaken. Tinidazole and atovaqoune were discontinued in November 2017, and ceftriaxone in December 2017. An oral treatment of 500 mg of Disulfiram per day was initiated in January 2018. Blood count, general chemistries, and urinalysis were obtained regularly and were satisfactory. The patient tolerated treatment, although he experienced profound fatigue, confining him mostly to bed on weeks five and six, during which time he later reported, “my mind was mush”. 

In mid-February 2018, at the end of the sixth week of treatment with disulfiram, the patient had a syncopal episode with concussion, requiring brief hospitalization. Disulfiram was discontinued. There was neither an intracranial bleed nor a cerebral contusion, and the patient recovered well. In the hospital, his imipramine and desipramine levels were elevated modestly above therapeutic range. A drug–drug interaction between disulfiram and imipramine was thought to be responsible for the syncopal episode, because when he was off disulfiram, his imipramine and desipramine levels were sub-therapeutic. 

Shortly thereafter, the patient was contacted, and he opined, “I think the treatment worked”. In late March 2018, in a telephone conversation, he advised, “I polished my shoes! Do you know how long it has been since I shined my shoes?” In consultation with his psychiatrist, he has been able to taper and/or discontinue much of his psychopharmacotherapy. Ketamine infusions were discontinued. Not only has he remained well, but he has progressively improved beyond the best status he had achieved when on extended intravenous antibiotic therapy and intensive psychopharmacotherapy. His depression has resolved significantly, and his memory and concentration have improved. He reported being able to do simple mathematics calculations “in my head”. He is requiring a normal amount of sleep, wakes refreshed, has normal energy, is active throughout the day, and is reading several newspapers ‘online’. He renewed his passport. He sustained a new bite June 2018, from a tick which he suspected was brought in to the home by one of his dogs. His primary care physician prescribed doxycycline, 100 mg PO BID, which the patient took for three weeks, although he queried whether or not disulfiram ought to be used instead. A dime-sized area of erythema and induration developed at the bite site, but no erythema migrans rash nor any obvious symptoms of illness suggested re-infection. He has remained well off all antimicrobial treatment thus far, until May 2019 (in excess of one year).

## 4. Case 3

This patient—a police officer, 35 years old at the time of his initial visit in October 1995—reported he had had a round, uniformly red rash on his upper inner right leg just below the knee, roughly the size of a lemon, some time 5–10 years earlier. It was thought to be a ‘spider bite’ and no treatment was given. He developed hip pain in 1992 with anxiety, panic attacks, and palpitations. In summer 1994, extreme fatigue developed, and by fall 1994 he was feverishness, with malaise and flu-like symptoms, chills, mild unrelenting headache, and a stiff neck. In December 1994, disorientation, memory and speech difficulties developed. 

He was active out of doors for work and recreation in Rockland County, New York. He hiked extensively in the lower Hudson River Valley and had an indoor/outdoor dog.

A physician treated him with amoxicillin for two weeks with improvement, but symptoms recurred following cessation of the drug. A second course of amoxicillin of four weeks duration likewise improved his symptoms, but they relapsed following cessation of the agent. The treating physician advised the patient that he suspected him to have Lyme disease and he was referred to an infectious diseases specialist, who did not agree with the diagnosis, as serologic tests were negative. Neurological consultation was obtained. An MRI of the brain with and without gadolinium was normal in May 1995, as was spinal fluid. Although Lyme disease was not able to be confirmed by laboratory methods, the neurologist advised the patient that it had not been ruled out. In June 1995 he was treated for six weeks with IV ceftriaxone, with initial intensification of joint symptoms, but with subsequent marked overall improvement. By fall of 1995 he experienced a relapse of symptoms, feeling ‘hung over’ and also a sense of breathlessness. A pulmonary evaluation was obtained. The angiotensin converting enzyme was elevated but pulmonary evaluation did not sustain a diagnosis of sarcoidosis.

The brain single photon emission computed tomography SPECT at Columbia Presbyterian Medical Center in November 1995 showed diffuse heterogeneous cerebral hypoperfusion, with decreased up-take in the white matter. A Lyme ELISA at the State University of New York at Stony Brook was negative, and a Western blot showed no bands on IgM but 39 and 41 kiloDalton (kDa) bands on IgG [53].

He experienced multi-system symptoms, including headaches, stiff neck, chills, photosensitivity, swollen glands that accompanied his cycling symptoms, mood swings, sleep disturbance, joint pain, debilitating fatigue, cognitive and balance difficulties, and diminished libido, among others. 

Aware of his cognitive difficulties, the patient often found himself in situations that could involve split-second decisions on resorting to deadly force. He sought and received medical leave from duty in the fall of 1995.

Formal detailed neuropsychological testing in May 1996 demonstrated impairments in practical reasoning, social judgment, auditory processing speed, primary visuospatial processing, and visual memory—areas of particular premorbid strength. These objective findings corroborated the patient’s subjective sense of cognitive difficulties and buttressed the administrative decision to grant medical leave.

From November 1995 to February 1996 he was treated with high dose amoxicillin (maximum dose was 3.5 g of amoxicillin Q 8 h, along with probenicid) aimed at CNS Lyme disease, achieving a peak amoxicillin blood level of 20.6 mcg/mL (therapeutic for CNS = 15–25 mcg/mL) [42]. Treatment was combined with clarithromycin in doses as high as 1.5 g PO Q 12 h. Despite this, he was getting lost while driving in familiar areas, had difficulty expressing himself, and experienced intolerable anxiety. Minocycline was substituted for clarithromycin.

In January 1996, a Lyme Western blot at the State University of New York at Stony Brook showed 30, 39, and 93 kiloDalton bands on IgM and 41, 45, and 58 kDa bands on IgG [53,54].

In March 1996, oral amoxicillin was discontinued, and IV ceftriaxone (2 g/day) was initiated and continued, with some hiatuses, until October 1996, when it was suspended due to the development of asymptomatic pseudocholelithiasis. Imipenem/cilastatin (1 g IV Q 8 h) was substituted until December 1996, and then cefotaxime (2 g IV Q 8 h) was administered until February 1997, when this drug was discontinued due to leukopenia. Oral minocycline was continued until October 1996, when clarithromycin was resumed and minocycline stopped. Altogether, treatment with IV antibiotics had continued for the bulk of the eleven months from March 1996 to February 1997, with occasional hiatuses.

During this treatment period, the patient reported improved mental sharpness, with easier word retrieval and facility of speech, diminished depression and anxiety, and improved sleep quality and libido.

Repeat brain SPECTs at Columbia Presbyterian Medical Center in October 1996 and April 1997 showed interval improvements, with progressive improvement in cortical heterogeneous and white matter hypoperfusion, however the pattern did not return to normality.

Oral antimicrobial therapy, with high dose amoxicillin and probenicid, combined with clarithromycin, was resumed in March 1997.

In April 1997, the patient was able to resume work on a limited schedule. The patient felt he had improved to about 65–70% of his normal ‘pre-morbid’ status and felt he was still making gains on oral antimicrobial therapy, although he experienced stiffness about his hips and deep bone pain in his thighs. Oral antibiotics were changed to cefuroxime (1 g) and azithromycin (0.5 g PO Q 12 h). 

In May 1997 at the State University of New York at Stony Brook, a Western blot showed 37 and 41 kDa bands in IgM and 18, 39, 41, and 93 kDa bands on IgG [53,54,55]. 

In late May 1997, the patient’s psychiatrist added donepezil, which resulted in improved word-finding and confidence with speech. The patient felt psychologically improved, and felt his condition was ‘holding’ on oral antimicrobial therapy.

In July 1997, on his own advice, he adopted a ‘pulse’ approach to oral antibiotic usage, partly due to loose bowels when on intensive treatment, despite ample probiotics. Diarrhea remitted. He felt at “75%” of his pre-morbid status and was able to resume full-time police work by fall 1997. Oral azithromcyin was discontinued and instead cefuroxime (2 g PO Q 12 h) was combined with doxycycline (150 mg PO Q 12 h) [56]. In December 1997, monotherapy with azithromycin (500 mg PO Q 12 h) was begun, to which cefuroxime was added in January 1998. He was maintaining his ground on oral antimicrobial therapy.

In February 1998, the patient sustained an Achilles tendon rupture while playing sports, and surgical repair was undertaken. He had never received any treatment with quinolone antimicrobials.

In July 1998, the patient was doing well on a rotating schedule of single antimicrobial agents, which included cefuroxime, doxycycline, amoxicillin, and clarithromycin, sequentially. In October 1998 the patient was feeling well and he suspended all antibiotic treatment. In April 1999, he was working hard, full time, occasionally doing ‘double shifts’. In July 1999, he noted spells of feeling ‘sick’ with malaise, sinusitis-like symptoms, stiffness about his hips, and some decrease in articulateness. In November 1999, he took a two-week course of left-over cefuroxime (1 g Q 12 h). In an exam in December 1999 he was noted to exhibit mild ataxia on tandem gait with eyes closed. Over the prior spring, summer, and fall, he had been hiking, including ‘bushwacking’, through dense brush in Harriman State Park and the Shawangunk Mountains—heavily tick-infested areas—but he was unaware of any known tick attachments. 

In February 2000, he reported experiencing monthly flares of symptoms, which had been absent since the prior spring but had begun to resurface in the summer and fall of 1999. High dose amoxicillin was resumed but then discontinued from March to August of 2000. Modest doses of amoxicillin were resumed until October 2000, then discontinued, as the patient was unclear as to the cause of his symptoms or the efficacy of amoxicillin. He modified his habits, with more rest, and avoidance of alcohol and caffeine. Nonetheless, right hip pain severe enough to interfere with sleep, dull head pressure, anxiety and panic attacks, and feeling ‘at a loss’ for words recurred. He suspected that Lyme disease was responsible. In January 2001 he resumed left-over doxycycline, using up to 400 mg/day, and noted that his hip pain resolved within a few days [56]. Further doxycycline was prescribed for a few months. 

In September 2001, he developed fatigue so severe he could barely get out of bed, along with muscle and joint pain and stiffness, and he resumed left-over doxycycline. By November 2001, he was feeling relatively well once again. He reported hiking in Harriman State Park several times per month until the fall of 2001, again, without known tick attachments.

In November 2001, a Western blot at the State University of New York at Stony Brook showed band 23 on IgM and bands 23 and 41 on IgG. 

Doxycycline was continued until January 2002, when it was discontinued by the patient, who was feeling relatively well. Within one month of discontinuing the drug, muscle and joint stiffness recurred. Doxycycline was resumed but after one additional month it did not seem to be conferring benefit. He had a deep aching in his thigh muscles ‘close to the bone’, diminished stamina, bilateral Achilles tendon pain, and recurrent cognitive symptoms. Treatment was changed to cefuroxime, combined with clarithromycin, and by June 2002 the patient was feeling remarkably better.

A decision was made to treat ‘episodically’ as seemed necessary, and discontinue when a satisfactory status was achieved, with the option to resume treatment if needed.

The patient remained relatively well off antibiotic therapy from June 2002 until January 2004, when sinusitis-like symptoms occurred. He was treated for sinusitis by a physician, with amoxicillin/clavulanate for 10 days, and felt better for several weeks. Sinusitis-like symptoms recurred, along with fatigue, leg pain, heaviness, stiffness, and weakness. He reconsulted the physician he had seen and was re-treated with the same regimen. All symptoms responded. The patient began to suspect Lyme disease once again because of the recurrence of this familiar constellation of symptoms. He used a friend’s left-over cefuroxime (1 g Q 12) for some three months and felt better.

He was feeling fairly well and able to function. He had had no known tick attachments but had been in the woods a lot with his dog at Harriman State Park, in June 2004. After the outing, he removed some 20 ticks from the animal. 

In late August 2004, off antimicrobials, a dry cough developed that lasted six weeks, which was resolved with cefuroxime but recurred 3 weeks later. This resolved with amoxicillin, which he acquired from a friend.

He commented that sinusitis had been a prominent symptom with the onset of his illness in 1994, and that an annoying ‘dry’ cough was a frequent symptom that responded to antibiotic therapy. He averred that on antibiotic therapy he generally would feel well, his anxiety would decrease, and his libido would improve. 

With recurring symptoms of Lyme disease, and ongoing exposure risk, a laboratory re-assessment for tick-borne infection was undertaken in December 2004, which revealed a positive Lyme ELISA at the State University of New York at Stony Brook, with an optical density of +0.206 and a positive cut-off of 0.152, but a concurrent Western blot showed only band 41 on IgM and IgG. Empiric re-treatment with cefuroxime and clarithromycin was instituted and continued until February 2005. With this, he felt better and suspended further treatment. 

In March 2005 right elbow pain and right knee pain developed, which required three–four weeks to resolve, and which flared again in early June 2005, along with a dry cough and some anxiety. He resumed left-over cefuroxime and clarithromycin—the joint pain improved within several days of resumption of antibiotics and the cough and anxiety resolved. 

In February 2006, he returned for care, reporting significant stiffness since the prior November, interfering with his functioning and quality of life. He was also experiencing loss of words and, once again, difficulty with his sense of direction, and expressed that he was living a ‘diminished’ life. Oral amoxicillin was prescribed at 3500 mg PO Q 12 h, and he remained on this between February and October 2006, feeling relatively well. 

He retired from the police department after 20 years of service in August 2006.

In late February 2007, he developed a mild fever and chills, worse with exertion, which lasted 10 days. Once again, he began to consider Lyme disease. During 2007, he resorted to short courses of azithromycin, doxycycline, or amoxicillin when symptoms which he interpreted as denoting Lyme disease (sinusitis, right knee and left Achilles tendon pain, anxiety) recurred, achieving temporary surcease and returning to a sense of relative well-being.

In late 2007 and early 2008, the patient experienced periodic ‘flares’ of symptoms he attributed to Lyme disease, with sinusitis-like symptoms, excessive somnolence, headache, feverishness, and a stiff neck. In May 2008, minocycline was prescribed.

His psychiatrist prescribed duloxetine, which was tried without benefit.

A Lyme ELISA at the State University of New York at Stony Brook, March 2008 was positive, with an optical density of 0.239 and a positive cut-off of 0.140, with a 41 kDa band on the IgM Western blot but no bands on the IgG blot. A Western blot at IGeneX in May 2008 showed the presence of 31, 34, and 41 kDa bands on the IgM Western blot [57].

A brain SPECT in April 2008 at Columbia Presbyterian Medical Center showed moderate global cortical hypoperfusion and heterogeneity, which had worsened compared to the prior study of April 1997.

In May 2008, otolaryngology was consulted in view of recurring sinusitis-like symptoms. An MRI of the sinuses was clear of any anatomical features typical of sinusitis, despite the patient’s symptoms.

In September 2008, despite treatment with minocycline, the patient reported a worsened cognitive status, with poor comprehension and sense of direction and an inability to read, which he usually enjoyed.

IV ceftriaxone was instituted in April 2009 and continued, with some hiatuses related to difficulties with vascular access during the spring and summer of 2009, for a total of 28 administered doses. Parenteral treatment was combined with oral clarithromycin. 

Around this time, serologies returned positive for exposure to *Babesia duncani* from several different laboratories, including the Public Health Laboratory of the County of Sonoma, California. Treatment aimed at babesia piroplasms was instituted for the first time, using atovaquone/proguanil. 

Parenteral and oral treatment was discontinued in September 2009, with greatly improved symptoms, including the resolution of cognitive difficulties, with the patient feeling generally well.

In November 2009, following a hike in Harriman State Park, he removed a fully engorged tick from his skin. A 50 cent piece-sized area of inflammation developed around the tick bite site, and a one-month course of doxycycline was taken.

In December 2009, chills, fatigue, and difficulty regulating his body temperature occurred, and the patient took left over amoxicillin, up to 5.25 g PO Q 12 h, with no impact on his chills. Ceftriaxone was resumed until March 2010 with atovaquone/proguanil, chills subsided, and he was feeling well.

In May 2010, he was on azithromycin and atovaquone liquid and feeling well with no further chills or sense of ‘air hunger’. Artemesinin was added (150 mg PO BID), and by September 2010 he reported feeling the best he had in a year [58].

From November 2011 to April 2014, the patient was under the care of a practitioner who utilized benzathine penicillin, atovaquone, clarithromycin, and sulfamethoxazole/trimethoprim for his treatment.

In November 2013, he had sustained a new, engorged deer tick attachment, for which doxycycline was prescribed for 21 days. 

In December 2013, serology for exposure to *Babesia duncani* was reactive from one commercial reference lab, and in March 2014 this was confirmed at a second commercial lab.

A Lyme PCR in whole blood was positive for detection of the OspA plasmid target, at IGeneX, April 2014, confirmed by southern dot blot. A Lyme Western blot at the same lab showed the presence of a 39 kDa band and faint bands at 31 and 41 kDa on IgM and 41 and 58 kDa bands on IgG, with a faint band at 23 kDa.

Benzathine penicillin was discontinued in May 2014, and the patient was treated with intramuscular ceftriaxone, initially at 1 g in a regimen of two days on and one day off (per the patient’s preference), along with a regimen of oral treatment mostly consisting of azithromycin (0.5–1 g PO QD), atovaquone (250 mg) and proguanil (100 mg PO X 2 BID), as well as liquid atovaquone (ranging from 750–2250, mg twice daily), but orals were usually limited to five consecutive days per week (again, per the patient’s preference). 

Due to inadequate control of symptoms, ceftriaxone was increased to 2 X 1 g IM, two days on and one day off, September 2014.

Treatment was changed to IV ceftriaxone between December 2015 and February 2016. At that point, an attempt to transition from parenteral ceftriaxone with oral azithromycin and atovaquone to oral minocycline or doxycycline with artemether/lumefantrine proved unsuccessful. All treatment was suspended in March 2016, but a return of the symptoms of tick-borne infection required reinstitution of atovaquone with azithromycin [58]. A fully implanted vascular access device was placed, and the patient was treated with IV ceftriaxone (2 g/day) from May 2016 until August 2016. From August 2016 to January 2018, the patient was treated with IM ceftriaxone (1 g X 2), two days on and one day off. Attempts at cessation of treatment with azithromcyin and atovaquone resulted in the onset of drenching night sweats.

The patient was advised of the impact of treatment with disulfiram in the patients described in Cases 1 and 2, and after due consideration, including a discussion of its potential risks and its uncertain utility in the treatment of human Lyme disease, he requested a trial of treatment with that agent. Correspondence was held with his treating psychiatrist before treatment was initiated. 

Disulfiram (250 mg tablets, 1–2 per day) was prescribed, and the patient initiated treatment in mid-January 2018, discontinuing all other antimicrobial treatment.

Disulfiram resulted in profound fatigue that interfered with functioning, and initially the patient was unable to tolerate more than 125 mg of disulfiram every other day. The dose was gradually increased from January to April, when he was able to tolerate 500 mg/day for about the last two months of treatment (ending in late May 2018). Periodic surveillance laboratory testing was satisfactory. Notable to the patient was that, despite the discontinuance of azithromycin and atovaquone in mid-January of 2018, he experienced no recurrence of night sweats, even shortly after initiating only low dose disulfiram. He remained clinically well on no antimicrobial treatment until December 2018. He noted improved libido following the course of disulfiram. In retrospect, he opined that the debilitating initial effects of disulfiram seemed most consistent with Jarisch–Herxheimer-like effects, which he had experienced with the application of conventional antibiotics during his course of care.

In November 2018, he requested interval testing for tick-borne diseases stating, however, that he remained clinically well. Surprisingly, Lyme PCR for detection of the plasmid target in the serum returned positive. *Babesia duncani* antibodies at the County of Sonoma Public Health Laboratory also returned reactive, at 1:256. *Babesia microti* antibodies were negative, as were direct detection for *B. microti* and *B. duncani* by PCR and F.I.S.H at IGeneX.

During December 2018 and January and February of 2019, the patient experienced left Achilles tendon and right hip pain, the latter interfering with sleep and reminiscent of past relapses of Lyme disease. On his own advice, he administered a five-day course of IM ceftriaxone, with left-over supplies that he had on hand. The course, from 12 February 2019 to 16 February 2019, reduced musculoskeletal pain by some 60%, by his estimation.

A second course of disulfiram was initiated on 19 February 2019, starting slowly with 0.5 X 250 mg every other day, with the intention to ramp the dose up over several weeks to 750 mg/day, remain at that dose for 90 days, and then discontinue the agent for a period of observation. 

## 5. Discussion

The persistence of borrelial and piroplasm infections despite treatment poses a dilemma for patients and physicians alike. Failure to treat may subject the patient to personal suffering, deterioration, and loss of function, and can eventuate in death [3,4,5,13,15,59,60,61,62,63,64,65,66] (Appendix A). Open-ended antimicrobial treatment is costly, requires medical oversight for safety, entails risks—including the risk of death—and theoretically risks the emergence of resistant strains of microbes, with public health implications [67,68].

Despite this conundrum, little attention and few resources have been directed towards the development of improved treatment methods for Lyme disease and babesiosis, with the goal of complete microbial eradication and clinical cure. Recently, however, at least four academic research centers have been focusing on the problem of borrelial persisters. Their initial approach has been to evaluate existing FDA-approved drugs in the United States Pharmacopeia for *in vitro* activity against borreliae, either singly or in various combinations [26,37,69,70,71,72,73]. Herbal preparations have also been evaluated for *in vitro* activity [74,75].

Disulfiram was first utilized as an industrial chemical for the vulcanization of rubber [76,77]. Factory workers exposed to the chemical were noted to have unpleasant reactions following the ingestion of alcohol. Subsequent studies confirmed this, and the agent was ‘repurposed’ to facilitate sobriety in persons with chronic alcoholism by aversive conditioning [78]. Although its effectiveness has been questioned, it has been in clinical use for some 60 years, and prescribed for extended periods of time in certain cases. 

Disulfiram metabolism is complex [79,80,81,82,83,84]. The drug and its metabolites, including carbon disulfide and diethyldithiocarbamate, have a wide distribution in mammalian tissues, including lipids and the central nervous system, prolonged half-lives, and may require one to two weeks for complete elimination after a dose. The accumulation of acetaldehyde through the blocking of aldehyde dehydrogenase is thought, in part, to be responsible for the disulfiram–ethanol reaction, but additional mechanisms may be operative [85].

Although generally safe, documented disulfiram adverse effects include encephalopathy [86,87], convulsion [88,89], cranial and peripheral neuropathy [90,91,92,93,94], toxic optic neuropathy [95], irreversible injury to the basal ganglia with permanent neurological deficits [96], hypertension [97], and drug-induced psychosis, presumably due to the inhibition of dopamine beta-hydroxylase [50,51,52]. Perhaps most concerning, disulfiram has caused liver injuries which have required liver transplantation and/or resulted in death [98,99,100,101,102,103,104].

Disulfiram can result in drug–drug interactions through the CYP450, CYP2E1, and other pathways, and can have a major impact on levels of other prescribed drugs, such as warfarin, phenytoin, barbiturates, opioids, tricyclic anti-depressants, hypo-glycemic agents, anti-histamines, benzodiazepines, CNS stimulants, and psychopharmacologic agents [76,77,85]. 

Careful monitoring, dosage adjustments, or a decision to defer disulfiram use in view of other concurrent therapies may be appropriate. Concurrent use of the nitro-imidazoles metronidazole and tinidazole is contraindicated. Alcohol ingestion must be avoided, and patients must be warned that some over-the-counter products may contain alcohol and must therefore be avoided. Disulfiram use should be avoided in persons with a known allergy to vulcanized rubber.

Disulfiram has been recognized to have a novel unanticipated utility for addictions other than alcohol [85], as an anti-cancer agent [105], and it reduced plaque-burden in a mouse model of Alzheimer’s disease [106]. 

At present, one can only speculate as to the mechanisms by which disulfiram appears to have conferred benefit in the three patients described herein. Its use was directed at chronic Lyme borreliosis, but it appears to have resulted in the remission of symptoms attributed not only to that infectious disease but, unexpectedly, of symptoms attributed to piroplasmosis.

When one considers the original industrial use of disulfiram in the vulcanization of rubber, and that natural latex rubber, as well as microbial cell membranes, consist primarily of lipids with an admixture of proteins, one can ponder whether or not disulfiram may affect microbial surface membrane flexibility and function.

*B. burgdorferi* has been shown to require zinc and manganese as co-factors for key biological processes [107,108]. It is possible that disulfiram’s high avidity for metal ions [85] may inhibit microbial metabolism. 

Disulfiram has been shown to have anti-mycobacterial properties [109] and to have good in vitro activity against multi-drug resistant *Staphylococcus aureus* [110]. Disulfiram-derived disulfides were found to have antibacterial properties [111]. 

Disulfiram has demonstrated anti-parasitic activity against giardia [112], malaria [113], leishmaniasis [114], and trypanosomiasis [115]. 

Detailed physico–chemical studies have suggested interdigitation of folded protein structures as a specific mechanism of the inhibitory action of disulfiram in giardiasis [112].

Any role for disulfiram in children, with their incompletely matured nervous systems, remains to be determined and should be approached with caution, as there is little experience with its use in children. Severe permanent neurological injury has been reported in accidental disulfiram poisoning in children [116,117]. On the other hand, it has been used, apparently safely, in adolescents (16–19 years of age) with alcohol addiction [118].

This report of treatment with disulfiram in three individuals at a dose and duration somewhat arbitrarily chosen depended upon patients’ subjectively reported experience. Nonetheless, the ability of two of these patients—each of whom had required and depended upon open-ended antimicrobial treatment in order to maintain a level of relative well-being and stability to discontinue all antimicrobial therapy, remain well, and experience further improvement in their status, over observation periods of 15 and 23 months—was a striking and unexpected outcome for them and for their treating clinicians. One patient relapsed at six months following completion of treatment and is being re-treated at a slightly higher dose for a more extended duration.

The experiences reported here resulted from the coalescence of work done by bench researchers, patient self-advocacy for use of an agent never previously utilized in humans for the treatment of tick-borne infection, and the discretion of the treating physician to utilize an FDA-approved agent for novel ‘off-label’ use. Nonetheless, this would not have been undertaken had not the patient of Case 1 and, subsequently, the patients of Cases 2 and 3, requested it. They must be regarded as pioneers.

Should studies by other investigators corroborate the benefit of disulfiram application in Lyme disease and babesiosis, then a formal study of disulfiram use in well-characterized persons with these infections, as well as those designated as suffering from Post-Treatment Lyme Disease Syndrome, should be considered. This study should use all known laboratory methods of evaluation, including antibody, cytokine, chemokine, proteomic, and metabolomic responses in serum and cerebrospinal fluid, as well as direct detection methods in bodily fluids and (when feasible) tissues, and neuroimaging and neuropsychological testing [119]. 

The putative anti-borrelial and anti-piroplasm mechanisms of the action of disulfiram, about which we have speculated, need to be clearly elucidated by appropriate scientific investigations. 

The optimal and sufficient dosage and duration of treatment need to be defined, because use of the lowest effective dose and shortest effective duration of disulfiram for these indications should serve to minimize adverse effects. The patients of Cases 1 and 2 weighed approximately 200 pounds and a disulfiram dose of 500 mg/day appears to have sufficed to maintain long term remission. The patient in Case 3 weighed some 220 pounds. Although a dose of 500 mg/day conferred enduring benefit, this dose did not prevent a clinical relapse of Lyme disease six months following completion of his course of therapy. As with most pharmacological agents, it is logical to assume that disulfiram may well require dose adjustments for weight, as suggested by the relapse which occurred in Case 3. Nonetheless, it will remain to be seen whether upward dosage adjustment and more prolonged duration will achieve enduring remission in Case 3—there may have been other reasons for treatment failure in his case beyond dosage-for-weight issues. 

Subsequent to the experiences reported here on the three initial individuals, disulfiram has been utilized in an additional two dozen carefully selected patients, who we do not report on in detail here. Results have been impressive, and disulfiram use, with careful supervision, has been relatively safe. Several individuals, whose rather precarious clinical status induced the use of very low doses of disulfiram (e.g., 125 mg every third day only), demonstrated dramatic improvement in their clinical status, which prompted a change in strategy, using disulfiram as a ‘maintenance’ therapy. For these patients, a decision was made to hold in abeyance increases in doses of disulfiram with a view towards inducing ‘remission’, as appears to have been achieved in Cases 1 and 2.

Hence, disulfiram as a single agent may offer an economical, practical, and relatively safe alternative to the use of combinations of long-term antibiotics and anti-parasitic therapies, which entail risk, expense, and inconvenience.

If a salubrious effect of disulfiram in the treatment of borreliosis and babesiosis is confirmed, then its utility in the treatment of other spirochetal diseases, including syphilis, relapsing fever, tropical treponematoses, leptospirosis, as well as certain parasitic diseases, should be explored.

It will remain to be seen whether disulfiram has a role in acute or early Lyme disease or babesiosis, and whether it is suitable for use for prophylaxis for tick-bites [120,121,122].

## 6. Conclusions

Disulfiram appears to have conferred benefit in the treatment of a limited number of patients with Lyme disease and babesiosis in a clinical setting. Formal controlled trials of this agent for the treatment of Lyme disease and babesiosis may be warranted along with elucidation of mechanisms of its apparent anti-borrelial and possible anti-babesial effects.

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
