# Peer review of "Disulfiram (Tetraethylthiuram Disulfide) in the Treatment of Lyme Disease and Babesiosis: Report of Experience in Three Cases"

_antibiotics, 2019, doi:10.3390/antibiotics8020072_

Round 1

Reviewer 1 Report

This manuscript, a case study of 3 patients treated with the FDA approved drug disulfiram. The use of this drug in treatment of complex chronic Lyme disease is new and the success reported by the author is tremendously important. I would anticipate that this publication will lead to significant and rapid translation to clinical practice.

The manuscript is well written, important and exceptionally well referenced.
I have 4 comments that might improve the presentation of the manuscript and a handful of typos/formatting errors.

Comment 1. A table giving demographics information on the 3 patients - ie demographics, a summary of years of illness, prior treatments (ie multiple abx, long duration), response to disulfiram, etc. would be useful in the results or discussion section for easy access for those without the time to read through the somewhat longer case study descriptions.

Comment 2. Introduction. ".... eliminated the need for further antimicrobial treatment during the period of observation.”  As one patient relapsed, this should be rephrased as two patients stably during the period of observation and one recovered for X period of time but relapsed.

Comment 3. Discussion: Attributing the replase in patient 3 to a ~ 20lb difference in weight seems highly speculative. This patient also had the least robust immune response. And genetic factors, both in the patient and the pathogens always need to be considered. There are many more possible variables other than a ~10% difference in weight so I would suggest phrasing this speculation more tentatively and/or suggesting other alternatives.

Comment 4. The explicit recognition of the value of patient-clinician-research partnerships and the respect and acknowledgement of the role of the patients as pioneers was wonderful. This is how clinical and biomedical research should be done. But so often is not. This was truly wonderful to read.

Typos/formatting glitches

Intro:

“It is known for the last few years many researchers are identifying novel compounds for Lyme disease treatment”

This sentence could be reworded for clarity. Possibly the first clause should be demarked by a comma or simplified.

“Disulfiram was identified as one of the top 20 hit molecules after screening 7450 drug molecules from different chemical libraries

Against what and how? (Borrelia and in vitro) – but this should be specified

“FDA approved”

Should be hyphenated

Case studies:

Late May 2008 shortly after having been on an overgrown bike trail in Setauket in Suffolk County, New York and having sat on a nearby pile of logs the patient not

Commas misplaced

Page 5/19 - tinidazone 500 mg. twice

Period not needed

“Once again, he began to consider Lyme.”

Lyme disease

“Benzathine penicillin was discontinued May 2014 and the patient was treated with

Intramuscular ceftriaxone”

Typo/formatting – capitalization.=

“During December 2018 and January and February of 2019 the patient experienced left Achilles tendon and right hip pain, the latter interfering with sleep. migratory and intermittent”

Typo/formatting

“On his own advice he administered a 5 day course of intramuscular ceftriaxone with left-over supplies that he had on hand 2/12/19-2/16/19 which reduced musculoskeletal pain by some 60% by his estimation.”

“A second course of disulfiram was initiated 2/19/19, starting slowly with..”

Date formatting inconsistent

Author Response

Antibiotics Manuscript ID: antibiotics-483236 Response to Reviewer 1 Comments

Comment 1: The reviewer’s suggestion to tabularize data of the three cases to facilitate readers’ apprehension of the cases at a ready glance in addition to the full written histories has merit. The author did endeavor to tabularize the data contained within the written presentation of Cases 1,2 and 3.   However, the complexity of clinical details, diagnostics and therapeutics playing out over many years in each of the cases and particularly in Case 3, decades, does not readily lend itself to tabularization.  Such tables themselves would be extremely lengthy and unwieldy. 

Also, the presentation of these cases falls well within what has come to be termed ‘narrative medicine’, e.g. the importance of ‘story-telling’: conveying the lived experiences of persons suffering from chronic illnesses and within marginalized populations.

See:

Charon R. Narrative medicine: A model for empathy, reflection, profession, and trust. JAMA. 2001 Oct 17;286(15):1897–902. DOI: https://doi.org/10.1001/jama.286.15.1897. [PubMed] [Google Scholar]

&

Charon R. From detached concern to empathy: Humanizing medical practice. Journal of Health Politics, Policy and Law. 2003;28(6):1121–5. [Google Scholar]

&

Muneeb A, Jawaid H, Khalid N, Mian A. The Art of Healing through Narrative Medicine in Clinical Practice: A Reflection. Perm J. 21:17-013. doi: 10.7812/TPP/17-013. Epub 2017 Oct 5. PubMed PMID: 29035184; PubMed Central PMCID: PMC5638639.

Comment 2: This comment has merit.  It should be pointed out that the manuscript was composed and essentially completed before additional critical information was developed November 2018.  The author agrees that this calls for modification of the introduction and also the title to reflect that Case 3 demonstrated evidence of relapse of Lyme disease (but not, apparently, babesiosis) despite experiencing a good remission over the first 6 months following treatment with disulfiram.

Comment 3: Although the author’s suspicion that the relapse in Case 3 may have been due to dosage for weight (10% greater than the weight of Cases 1 & 2) other mechanisms might be operative that could explain the relapse as pointed out by the reviewer.  The suggestion to be more ‘tentative’ about whether the dosage was inadequate versus other incompletely defined factors/reasons is valid and the text will be modified along the lines suggested by the reviewer.

Comment 4: The author appreciates this comment.  This trial of treatment in the 3 cases described would never have been undertaken or suggested to any patient by the practitioner on the practitioner’s initiative.  It was only agreed to and  implemented following the earnest request of the patient of  Case 1.  The author/clinician informed himself about this already FDA-approved agent and the pertinent recent research as to its in vitro activity against borrelia.  The potential risks of treatment were discussed with the patient as well as the unknown utility of the drug in the treatment of borrelial infection in humans.  Treatment was provided with frequent laboratory monitoring and communication with the patient.  The unexpectedly dramatically positive response encouraged the clinician to share this singular experience selectively with the patients of Cases 2 & 3.  Each was informed to the best of the clinician’s ability of the potential risks, potential benefits as well as the uncertainties involved in the use of disulfiram for the treatment of Lyme disease and each requested to receive a trial of treatment with the agent.  Other physicians involved in the patients’ care were also consulted.  As stated, the author regards the patients of Cases 1-3 as ‘Pioneers’ deserving of recognition (although they are not seeking recognition).

Regarding typographical, formatting and syntax errors, the author appreciates the attention to detail given by Reviewer 1 and each of the instances cited will be corrected or revised.

Reviewer 2 Report

This is a manuscript dealing with possibly Borrelia- or Babesia- associated post-infectious possibly autoimmune-mediated acute and chronic inflammatory symptoms in possible Lyme borreliosis cases. The clinical case definitions remain cryptic and do not follow European or US guidelines. There is no disease such as chronic relapsing neurologic Lyme disease and babesiosis. The paper and the underlying disease of the patients remain unclear and the author´s diagnoses are highly speculative. The treatment schedules and the choice of substances must be seen as off label and highly critical for the patients. Neither the clinical study standards nor the presentation of clinical data correspond to the standards of good clinical study practice. As such the diagnoses and the suggested so called treatment success remain highly doubtable because of possible confounders and simple misdiagnosis. According to the data presented it is even not clear how the diagnosis of LB and babesiosis in the study patients was achieved. Most importantly the choice of substances and the treatment regimens are not based on any sound scientific data. As such the presentation of data does not necessarily support the results as presented in the paper and does not meet the standards for publication in a renowned scientific journal!

Author Response

Antibiotics Manuscript ID: antibiotics-483236 Response to Reviewer 2 Comments

I appreciate that Reviewer 2 found my English language and style satisfactory.

Reviewer 2 apparently completely disapproved of the entire manuscript.  The introduction, background  and references, research design, methods, results and conclusions are all checked off as “Not applicable” which the author takes to mean that Reviewer  2 found every element of the manuscript invalid. 

Comments and Suggestions for Authors of Reviewer 2 clarify that Reviewer 2 is highly dubious of the diagnoses in the three cases presented, disbelieving in toto of the possibility of chronic relapsing Lyme disease as well as of chronic relapsing babesiosis and incredulous of and offended by the nature of the treatments which the patients had received during the courses of their illnesses as well as dubious as to any conclusions that may be able to be drawn as to the effect of application of disulfiram in their cases.

The position of Reviewer 2 adheres closely to that propounded by the Infectious Diseases Society of America, reinforced by the Centers for Disease Control and Prevention of the United States of America.

Certainly one can take this position if one chooses, however there exists a very substantial body of peer-reviewed literature developed over decades by highly credible scientists and physicians worldwide that demonstrates the reality of chronic persistent neurologic Lyme disease despite application of putatively curative antibiotic therapy.  Many hundreds of such references exist.  I have cited a sufficient number of them in the references of the manuscript.

The designation Post-treatment Lyme Disease Syndrome (PTLDS) was proposed to categorize that subset of individuals diagnosed with and treated for Lyme disease who remain ill.  PTLDS originally was agnostic as to the cause of residual symptoms, hypothesizing equally the possibility of chronic persistent infection as the cause, versus a truly ‘post-infectious’ immune derangement or autoimmune condition triggered by a no-longer-present infection by B.burgdorferi.  Both hypotheses are legitimate theoretical concepts that respected physicians and scientists entertain.

Many clinicians responsible for persons with Lyme disease and other tick-borne diseases have witnessed the personal suffering, deterioration and sometimes death that ensues when treatment with antimicrobials is withheld from persons experiencing relapse of symptoms despite receipt of ‘recommended’ regimens of antibiotics.

So, this represents a real dilemma for both patient and physician.  What is one to do?  Due to the dogmatic position taken by many in the medical profession, including CDC in the United States and the IDSA, patients have no options for treatment other than merely symptomatic therapies.  Provision of antimicrobials during the course of care of the patients described in Cases 1-3 was not because either the patient or the physician wanted to treat with these agents: it was because they had to be treated in order to allow a reasonable quality of life.  There was no other option, unless the patients were to be abandoned to their illnesses.  So, it is a very unsatisfactory situation.  I acknowledge this.  I share Reviewer 2’s dismay in the review of these cases. 

Babesiosis is not  uncommon as a co-infection found simultaneously along with Lyme disease in patients. It is also increasingly being recognized as difficult to eradicate despite application of the best available anti-piroplasm treatments known to man.  This situation is exemplified in the three cases.  Their symptoms of babesiosis relapsed when specific treatment aimed at piroplasms was withheld.  Similar logic applies: withhold treatment an allow the disease to manifest itself, or treat, realizing that the treatment is suppressive and not fully eradicative.

Regarding the question of accuracy of diagnosis of the three patients, raised by Reviewer 3, I would like to point out a number of issues.  The laboratories utilized for the diagnosis of babesiosis include a county public health laboratory (Sonoma County, California).  This laboratory has been involved in the discovery of what was then called WA-1, as the index case was found in a Washington State man.  Later, the illness was renamed Babesia duncani.  No one has questioned the credibility of that laboratory or the validity of its results. 

The New York State Department of Health evaluates all private laboratories utilized by New York State physicians for New York State residents and applies extremely stringent requirements to assure that the methods utilized are valid.  IGeneX, at which several of the patients have had detection of the DNA of the Lyme organism by PCR methodology and the RNA of the babesia organism by F.I.S.H. methodology have all been extensively ‘vetted’ and approved by NYS DOH.  This ought to give reasonable comfort that the results are meaningful and not meaningless.

Still, everything must be taken in the context of the patients’ histories and physical findings.  The persons described in the manuscript have experienced complex and evolving multi-system illness over time.

Each has had very extensive epidemiologic exposure risk and known tick attachments in some of the most highly hyper-endemic regions of the United States with highest incidence of Lyme disease and babesiosis anywhere in the world.

The author finds it puzzling that the diagnosis of central nervous system Lyme disease is doubted for Case 2, in particular.  This patient has ‘bullet-proof’ evidence of central nervous system Lyme disease that satisfies all academic criteria:  positive LymeELISAs, positive IgG Lyme Western blots, selective intra-thecal antibody synthesis with rising CSF/serum indices, positive Lyme PCRs, and clinical features of neurologic involvement which worsened when treatment was withheld and improved when treatment was applied.

I would encourage Reviewer 2 to look again at the items appended in Supplementary 1 and Supplementary 2a.  These include positive spinal fluid culture for isolation of B. burgdorferi from the patient described in Case 1 of that article.  The culture was performed at the Centers for Disease Control in Fort Collins, Colorado, U.S.A. from my patient who had previously been treated with 21 days of intravenous cefotaxime and four months of oral minocycline.  She had been ill for many years with an unexplained chronic meningoencephalitis with lymphocytic pleocytosis and had been seronegative for the first several years of her illness.  So this was a case of culture proven central nervous system borreliosis which had relapsed despite prior putatively curative treatment and who had been seronegative.  She, like the patients described in the current manuscript with Antibiotics, had pursued a puzzling and tortuous course with a host of odd manifestations but was ultimately proven to be a case of chronic relapsing culture proven antibiotic treatment failure.

The author is well trained with a background and training in surgical critical care medicine and anatomic pathology and follows his patients with a rigorous, discerning and scientific approach.  The author has collaborated with researchers and scientists of the first rank both domestically within the United States and internationally.  The author has extensive experience caring for persons with tick-borne diseases in the lower Hudson Valley, an area of the world with one of the highest incidences of Lyme disease.

I respect right of Reviewer 2 to maintain his or her opinion.

I would hope that the Cases presented in the current manuscript might serve to stimulate the curiosity of Reviewer 2 as to the lived experiences of these very ill individuals as they and their clinicians grapple with how to secure an acceptable quality of life in the face of what many scientists and clinicians have found to be formidable tick-borne infections, not so easily vanquished as is commonly claimed.

Many academicians insist that ‘true’ Lyme disease is adequately represented by that small ‘apex’ of what others believe is a large pyramid of affected persons .  That ‘apex’ is limited only to those who have a known deer-tick (Ixodes scapularis) bite occurring in a proven Lyme-endemic area, have a classic bull’s eye rash and who resolve their illnesses completely with a short course of antibiotics of no more than 30 days duration.  Persons having symptoms recur despite such treatment are then classified as PTLDS, but with the added (and unjustified) ‘spin’ that this is strictly a post-infectious condition due only to an  infection-triggered (but no longer driven by persistent infection)  autoimmune or hyper-inflammatory condition.

Frontline clinicians, confronted with actual persons suffering from Lyme disease and following them over extended periods of time are left to deal with the broader-based pyramid of infected individuals who often display a much wider array of symptoms, variable immune responses to the infectious agent – including seronegative cases and those with less than fully diagnostic Lyme Western blots - who nonetheless often display bands of high specificity which denote exposure to the B. burgdorferi.

Although Reviewer 2 is entitled to be skeptical (and the author also believes a ‘healthy skeptiicsm’ is an appropriate stance for both physicians and scientists) the author can assure Reviewer 2 that he (the author) has a very high level of confidence in the correctness of the diagnoses of chronic relapsing neuroborreliosis and chronic relapsing babesiosis in each of the 3 patients described in Cases 1,2 and 3.  The author believes this is supported by adequate laboratory data and also by the response to application of treatment aimed selectively at both Lyme borreliosis and babesiosis and relapse of symptoms characteristic of each illness with withholding of therapy and resolution when treatment was resumed.

I append a sampling of references by multiple authors speaking to the bona fide nature of chronic & neurologic Lyme disease and chronic persistent infection despite application of ‘appropriate’ antibiotic therapy.

This highlights the urgent need for the development of improved methods of treatment for both Lyme disease and for babesiosis.  The clinical observations concerning disulfiram’s apparent utility which, after all, ARE merely anecdotal and not in the setting of a formal, organized academic study, nonetheless were dramatic and unexpected and this impelled the author to go to the considerable lengths and effort to prepare a manuscript to bring to the attention of others in the medical and scientific communities that this merits further formal evaluation.

In that regard, I would also like to bring to the attention of Reviewer 2 and the editors of MDPI-Antibiotics that the experiences reported by the author and shared selectively with colleagues has inspired, at least in part, a formal blinded study of disulfiram application in persons with well-characterized Lyme disease with Brian A, Fallon, M.D.,M.P.H. as Principal Investigator.  Please refer to clinicaltrials.gov and see NCT03891667 which is anticipated to begin sometime in 2019.

Finally, it is important to appreciate that there is now widespread recognition of the limitations of many of the assumptions from prior decades about the nature of Lyme disease and other tick-borne diseases and also of the deficiencies of the two-tier system of testing for Lyme disease and the need for vastly improved methods of diagnosis.  The United States Health & Human Services Tick-borne Diseases Working Group (HHS TBDWG) has issued preliminary reports which outline the extensive scope of work that is necessary to move forward with diagnosis, treatment and control of tick-borne infectious diseases. These reports can be found at

https://www.hhs.gov/ash/advisory-committees/tickbornedisease/reports/index.html

REFERENCES: CHRONIC PERSISTENT INFECTION DESPITE INTENSIVE                                                       ANTIBIOTIC TREATMENT

1              Preac-Mursic V, Weber K, Pfister HW, Wilske B, Gross B, Baumann A, Prokop J. Survival of Borrelia burgdorferi in Antibiotically Treated Patients with Lyme borreliosis. Infection 1989;17:355-359.

2              Preac-Mursic V, Pfister HW, Spiegel H, Burk R, Wilske B, Reinhardt S, Bohmer R. First Isolation of Borrelia burgdorferi from an Iris Biopsy. J Clin Neuro-ophthalmol 1993;13:155-161.

3              Haupl T, Hahn G, Rittig M, Krause A, Schoerner C, Schonherr U, Kalden JR, Burmester GR. Persistence of Borrelia burgdorferi in Ligamentous Tissue from a patient with Chronic Lyme borreliosis. Arthritis & Rheumatism 1993;36:1621-1626.

4              Lawrence C, Lipton RB, Lowy FD, Coyle PK. Seronegative Chronic Relapsing Neuroborreliosis. Eur Neurol 1995;35:113-117.

5*** Liegner KB. Lyme Disease: The Sensible Pursuit of Answers.(Guest Commentary). J Clin Microbiol 1993;31:1961-1963.

6              Liegner KB. B. burgdorferi - Seek and Ye Shall Find. Expanding the Envelope. (Guest Editorial). J Spirochetal and Tick-borne Dis 1994;1:79-81.

7              Liegner KB. Treatment of Late Lyme Disease: a Challenge to Accept. (Reply to Letter). J Clin Microbiol 1994;32:1416.

8***       Liegner KB, Rosenkilde CE, Campbell GL, Quan TJ, Dennis DT. Culture-confirmed treatment failure of cefotaxime and minocycline in a case of Lyme meningoencephalomyelitis in the United States. In: Program and abstracts of the Fifth International Conference on Lyme Borreliosis, Arlington, Va., May 30-June 2, 1992. Bethesda, Md.:Federation of American Societies for Experimental Biology, 1992:A11.

9              Liegner KB, Agricola MD, Bayer ME, Duray PH. Chronic Lyme disease (CLD): A Costly Dilemma. Program and Abstracts. Sixth International Conference on Lyme Borreliosis. Bologna, Italy. June 19-22, 1994. Abstract P012M

10           Liegner KB, Ziska M, Agricola MD, Hubbard JD, Klempner MS,  Coyle PK, Bayer ME, Duray PH. Fatal Chronic Meningoencephalo-myelitis (CMEM) With Massive Hydrocephalus, In A New York State Patient With Evidence Of Borrelia burgdorferi (Bb) Exposure. In: Program and Abstracts. Sixth International Conference on Lyme Borreliosis. Bologna, Italy. June 19-22, 1994. Abstract P041T.

11           Liegner KB, Shapiro JR, Ramsay D, Halperin AJ, Hogrefe W, Kong L.  Recurrent erythema migrans despite extended antibiotic treatment with minocycline in a patient with persisting Borrelia burgdorferi infection. J Amer Acad Derm 1993;28:312-4.

12           Lopez-Andreu JA, Ferris J, Canosa CA, Sala-Lizarraga JV. Treatment of Late Lyme Disease: a Challenge to Accept. (Letter). J Clin Microbiol 1994;32:1415-1416.

13           Nocton JJ, Dressler F, Rutledge BJ, Rys PN, Persing DH, Steere AC. Detection of Borrelia burgdorferi DNA by Polymerase Chain Reaction in Synovial Fluid From Patients With Lyme Arthritis. N Engl J Med 1994;330:229-234.

14           NIAMSD & NIAID Clinical Courier. Vol. 9, No. 5 August 1991. ISSN 0264-6684. Diagnosis and Treatment of Lyme Disease.

REFERENCES: CHRONIC AND NEUROLOGIC LYME DISEASE                                              (INCLUDING FATAL CASES)

                Straubinger RK. PCR-Based Quantification of Borrelia burgdorferi Organisms in Canine Tissues over a 500-Day Postinfection Period. J Clin Microbiol 2000;38:2191-2199.

                Liegner KB. Lyme Disease: The Sensible Pursuit of Answers (Guest Commentary). J Clin Microbiol 31:1961-1963, 1993.

                Weder B, Wiedersheim P, Matter L, Steck A, Otto F. Chronic progressive neurological involvement in Borrelia burgdorferi infection. J Neurology 1987;234:40-43.

                Ackermann R, Gollmer E, Rehse-Kupper B. Progressive Borrelien-Enzephalomyelitis.  Chronische Manifestation der Erythema-migrans Krankheit am Nervensystem. Dtsh. Med. Wochenschr.110(26)(1985)1039-1042.

                Preac-Mursic V, Weber K, Pfister HW, Wilske B, Gross B, Baumann A, Prokop J. Survival of Borrelia burgdorferi in Antibiotically Treated Patients with Lyme borreliosis. Infection 1989;17:355-359.

                Liegner KB, Shapiro JR, Ramsay D, Halperin AJ, Hogrefe W, Kong L.  Recurrent erythema migrans despite extended antibiotic treatment with minocycline in a patient with persisting Borrelia burgdorferi infection. J Amer Acad Derm 1993;28:312-4.

                Lawrence C, Lipton RB, Lowy FD, Coyle PK. Seronegative Chronic Relapsing Neuroborreliosis. Eur Neurol 1995;35:113-117.

***         Liegner KB, Duray P, Agricola M, Rosenkilde C, Yannuzzi L, Ziska M, Tilton R, Hulinska D, Hubbard J, Fallon B. Lyme Disease and the Clinical Spectrum of Antibiotic-Responsive Chronic Meningoencephalomyelitides. J Spirochetal and Tick-borne Dis 1997;4:61-73.

                Liegner KB & Jones CR.  Fatal progressive encephalitis following an untreated deer tick attachment in a 7 year-old Fairfield County, Connecticut child. [Abstract] VIII International Conference on Lyme Disease and other Emerging Tick-borne Diseases, Munich, Germany, June 1999.

                Fallon BA, Tager F, Fein L, Liegner K, Keilp J, Weiss N, Liebowitz MR. Repeated Antibiotic Treatment in Chronic Lyme Disease. J Spirochetal and Tick-borne Dis 1999;6:94-102.

                Miklossy J, Kuntzer T, Bogousslavsky J, Regli F, Janzer RC. Meningovascular form of neuroborreliosis: Similarities between neuropathological findings in a case of Lyme disease and those occurring in tertiary Neurosyphilis. Acta Neuro Pathol 1990;80:568-572.

                Oksi J, Uksila J, Marjamaki M, Nikoskelainen J, Viljanen MK. Antibodies against whole sonicated Borrelia burgdorferi spirochetes, 41-kilodalton flagellin, and P39 protein in patients with PCR- or culture-proven late Lyme borreliosis. J Clin Microbiol 1995;33:2304-15.

                Bertrand E, Szpak GM, Pilkowski E, Habib N, Lipczynska-Lojkowska W, Rudnicka A, Tylewska-Wierzbanowska S, Kulczycki J. Central Nervous System Infection Caused by Borrelia burgdorferi. Clinico-Pathological Correlation of Three Post-Mortem Cases. Folia Neuropathol 1999;37:43-51.

                Haass Anton. Lyme Neuroborreliosis. Current Opinion in Neurology. 1998;11:253-258.

                Kohler J, Kern U, Kasper J, Rhese-Kupper B, Thoden U. Chronic central nervous system involvement in Lyme borreliosis. Neurology 1988;38:863-867.

Kollikowski HH, Schwendemann G, Schulz M, Wilhelm H, Lehmann HJ. Chronic borrelia encephalomyeloradiculitis with severe mental disturbance: immunosuppressive versus antibiotic therapy. J Neurol 1988;235:140-142.

                Petrovic M, Vogelaers D, Van Renterghem L, De Reuck J, Afschrift M. Lyme Borreliosis - A Review of the Late Stages and Treatment of Four Cases. Acta Clinica Belgica 1998;53-3:178-1+83.

                Straubinger RK, Summers BA, Chang Y-F, Appel MJG. Persistence of Borrelia burdgorferi in Experimentally Infected Dogs after Antibiotic Treatment. J Clin Microbiol 1997;35:111-116.

                Liegner KB, Rosenkilde CE, Campbell GL, Quan TJ, Dennis DT. Culture-confirmed Treatment Failure of Cefotaxime and Minocycline in a Case of Lyme Meningoencephalomyelitis in the United States. Program and Abstracts. V International Conference on Lyme Borreliosis. Abstr. 63 P. A11, Arlington, VA. May/June 1992.

                Omasits M, Seiser A, Brainin M. Zur rezidivierenden und schubhaft verlaufenden Borreliose des Nervensystems. Wiener clinische Wochenschrift1990;102:4-12.

                Liegner KB, Ziska M, Agricola MD, Hubbard JD, Klempner MS, Coyle PK, Bayer ME, Duray PH. Fatal Chronic Meningoencephalomyelitis (CMEM) With Massive Hydrocephalus, In A New York State Patient With Evidence of Borrelia Burgdorferi Exposure. Program and Abstracts, VI International Conference on Lyme Borreliosis. Abstr. P041T. Bologna, Italy, June 19-22, 1994.

                Merlo A, Weder B, Ketz E, Matter L. Locked-in state in Borrelia burgdorferi meningitis. J Neurol 1989;236:305-306.

                Oksi J, Kalimo H, Marttila RJ, Marjamaki M, Sonninen P, Nikoskelainen J, Viljanen MK. Inflammatory brain changes in Lyme borreliosis. A report on three patients and review of literature. Brain 1996;119:2143-2154.

Wokke JHJ, van Gijn J, Elderson A, Stanek G. Chronic forms of Borrelia burgdorferi infection of the nervous system. Neurology 1987;37:1031-1034.

                Fallon BA, Kochevar JM, Gaito A, Nields JA. The Underdiagnosis of Neuropsychiatric Lyme Disease in Children and Adults. Psychiat Clin NA 1998;21:693-703.

Hodzic et.al. Persistence of Borrelia burgdorferi following Antibiotic Treatment in Mice. Antimicrobial Agents & Chemotherapy. May 2008, Vol. 52(5)1728-36.

                Livengood JA & Gilmore RD Jr. Invasion of human neuronal and glial cells by an infectious strain of Borrelia burgdorferi. Microbes and Infection. 2006

Additional pertinent references:

                Duray P. Capturing Spirochetes from Humans. (Editorial) American Journal of Clinical Pathology 1993;99:4-6.

                Schmidli J, Hunziker T, Moesli P, Schaad UB. Cultivation of Borrelia burgdorferi from Joint Fluid Three Months After Treatment of Facial Palsy Due to Lyme Borreliosis. (Letter). J Infect Dis 1988;158:905-6.

                Preac Mursic V, Marget W, Busch U, Pleterski Rigler D, Hagl S. Kill Kinetics of Borrelia burgdorferi and Bacterial Findings in Relation to the Treatment of Lyme Borreliosis. Infection 1996;24:9-16.

                Liegner KB, Garon G, Dorward D. Lyme Borreliosis (LB) Studied with the Rocky Mountain Laboratory (RML) Antigen Capture Assay in Urine. In: Program and abstracts of the Fifth International Conference on Lyme Borreliosis, Arlington, Va., May 30-June 2, 1992. Bethesda, Md.:Federation of American Societies for Experimental Biology, 1992:A 18.

                Oksi J, Uksila J, Marjamaki M, Nikoskelainen J, Viljanen MK.Antibodies against whole sonicated Borrelia burgdorferi spirochetes, 41-kilodalton flagellin, and P39 protein in patients with PCR- or culture-proven late Lyme borreliosis. J Clin Microbiol 1995;33:2304-15.

                Dorward DW, Fischer ER. Virulent Borrelia burgdorferi specifically attach to, activate, and kill TIB-215 Human B-lymphocytes. (Abstract). VIII Annual Lyme Disease International Scientific Conference on Lyme Borreliosis and other Spirochetal and Tick-borne Diseases. Vancouver, BC. April 28,29, 1995.

                Georgilis K, Peacocke M, Klempner MS. Fibroblasts Protect the Lyme Disease Spirochete, Borrelia burgdorferi, from Ceftriaxone In Vitro. J Infect Dis 1992;166:440-4.

                Klempner MS, Noring R, Rogers RA. Invasion of Human Skin Fibroblasts by the Lyme Disease Spirochete, Borrelia burgdorferi. J Infect Dis 1993;167:1074-81.

                Preac Mursic V, Wanner G, Reinhardt S, Wilske B, Busch U, Marget W. Formation and Cultivation of Borrelia burgdorferi Spheroplast-L-Form Variants. Infection 1996;24:218-226.

                Liegner KB. Chronic Persistent Infection and Chronic Persistent Denial of Chronic Persistent Infection in Lyme Disease. Address given at the 6th International Conference on Lyme Disease and Other Tick-borne Diseases, Atlantic City, NJ, May 5 & 6th, 1993.

                Priem S, Wolbart K, Rittig MG, Burmester GR et. al. Detection of Borrelia burgdorferi by PCR in Synovial Membrane, but Not in Synovial Fluid in Patients with Lyme Arthritis. (Abstract#D661). Proceedings VII International Congress on Lyme Borreliosis. June 16-21, 1996, San Francisco, CA.

                Dattwyler RJ, Volkman DJ, Luft BJ, Halperin JJ, Thomas J, Golightly MG. Seronegative Lyme Disease. Dissociation of T- and B-Lymphocyte Responses to Borrelia burgdorferi. N Engl J Med 1988;319:1441-6

                Schutzer SE, Coyle PK, Belman AL, Golightly MG, Drulle J. Sequestration of antibody to Borrelia burgdorferi in immune complexes in seronegative Lyme disease. Lancet 1990;335:312-15.

                Dorward DW, Fischer ER. Virulent Borrelia burgdorferi specifically attach to, activate, and kill TIB-215 Human B-lymphocytes. (Abstract). VIII Annual Lyme Disease International Scientific Conference on Lyme Borreliosis and other Spirochetal and Tick-borne Diseases. Vancouver, BC. April 28,29, 1995.

                Georgilis K, Peacocke M, Klempner MS. Fibroblasts Protect the Lyme Disease Spirochete, Borrelia burgdorferi, from Ceftriaxone In Vitro. J Infect Dis 1992;166:440-4.

                Klempner MS, Noring R, Rogers RA. Invasion of Human Skin Fibroblasts by the Lyme Disease Spirochete, Borrelia burgdorferi. J Infect Dis 1993;167:1074-81.

                Preac Mursic V, Wanner G, Reinhardt S, Wilske B, Busch U, Marget W. Formation and Cultivation of Borrelia burgdorferi Spheroplast-L-Form Variants. Infection 1996;24:218-226.

                Schmidli J, Hunziker T, Moesli P, Schaad UB. Cultivation of Borrelia burgdorferi from Joint Fluid Three Months After Treatment of Facial Palsy Due to Lyme Borreliosis. (Letter). J Infect Dis 1988;158:905-6.

                Preac Mursic V, Marget W, Busch U, Pleterski Rigler D, Hagl S. Kill Kinetics of Borrelia burgdorferi and Bacterial Findings in Relation to the Treatment of Lyme Borreliosis. Infection 1996;24:9-16.

                Coyle PK, Deng Z, Schutzer SE, Belman AL, Benach J, Krupp LB, Luft B. Detection of Borrelia burgdorferi antigens in cerebrospinal fluid. Neurology 1993;43:1093-1097.

                Coyle PK, Schutzer SE, Deng Z, Krupp LB, Belman AL, Benach JL, Luft BJ. Detection of Borrelia burgdorferi-specific antigen in antibody-negative cerebrospinal fluid in neurologic Lyme disease. Neurology 1995;45:2010-5.

                Girschick HJ, Huppertz HI, Russmann H, et al. Intracellular persistence of Borrelia burgdorferi in human synovial cells. Rheumatol Int (1996)16:125-132.

                Chary-Valckenaere I, Jaulhac B, Champigneulle J et al. Ultrastructural Demonstration of Intracellular Localization of Borrelia burgdorferi in Lyme Arthritis.(Letter). Br J Rheumatol (1998)37:468-470.

                Staubinger RK, Summers BA, Chang Y, Appel MJG. Persistence of Borrelia burgdorferi in Experimentally Infected Dogs after Antibiotic Treatment. J Clin Microbiol 1997;35:111-116.

                Brorson O, Brorson SH. Transformation of cystic forms of Borrelia burgdorferi to normal mobile spirochetes. Infection 1997;25:240-246.

Ma Y, Weis JJ. Borrelia burgdorferi Outer Surface Lipoproteins OspA and OspB Possess B-Cell Mitogenic and Cytokine-stimulatory Properties. Infection and Immunity 1993;61:3843-3853.

Livengood JA, Gilmore RD Jr. Invasion of human neuronal and glial cells by an infectious strain of Borrelia burgdorferi. Microbes and Infection xx(2006)1-9.

Embers ME, Barthold SW et.al. Persistence of Borrelia burgdorferi in Rhesus Macaques following Antibiotic Treatment of Disseminated Infection. PLoS One 7(1):e29914.

Reviewer 3 Report

Important article with many much-needed and well researched references; good detail in the case presentations- makes the report lengthy but in my opinion I prefer this level of detail.

Minor points:

In the introduction, rationale for use of disulfiram in treating Borrelia is appropriately mentioned, but not for its use in babesiosis- however the body of the article does mention, as do references, its potential as an antimicrobial agent for other parasites and this should be added to the introduction

Over-uses/misuses the word "some" in describing passage of time.

While it is mentioned in each case that the unusual therapeutic option of using disulfiram was discussed with the respective patients, to keep this paper consistent with currently recommended usage, the buzzwords "informed consent" should be added.

References are provided that mention persistence of infection in immunocompromized patients, and here the patient was shown to be IgG deficient. However, the connection between this patient's infection persistence and his immune deficiency is absent and should be pointed out.

Page 12- speculation about "vulcanization" of the microbial membrane- pure speculation? If no reasonable references or a better description of this theory can be included, then I suggest it be removed.

The extensive antibiotic usage and high doses prescribed in each case may be subject to criticism- so be it. However, I prefer adding comments to emphasize that despite these aggressive regimens of potent, high doses and combinations of antibiotics, the infections apparently persisted- making the demonstrated efficacy of a relatively short course of single agent disulfiram all the more remarkable

Author Response

Antibiotics Manuscript ID: antibiotics-483236 Response to Reviewer 3 Comments

I appreciate the overall favorable estimation of the manuscript by Reviewer 3 and I acknowledge the constructive criticisms levied in the reviewers section, Minor Points and I will take these in to consideration in revision of the manuscript.

The effect of disulfiram on babesiosis was only became apparent in following the courses of Cases 1,2 & 3 and was in no way anticipated by the author when undertaking these trials of treatment.  The issue of effect of disulfiram on babesiosis is mentioned in the discussion section, but not in the Introduction where such mention does indeed seem not inappropriate.  This will be incorporated in the revision.

Attention to over-use of ‘some’ in relation to passage of time will be given, with elimination or clarification of time duration/sequences.

The patient of Case 1, in particular, had documented IgG deficiency, and although this is mentioned during the case presentation, this issue (e.g. variability of immune competence as it may affect thel outcomes of individuals affected by Lyme disease) is worth emphasizing and this will be added to the discussion

The author agrees with Reviewer 3 that ‘vulcanization’ of microbial cell membrane and how this might affect function is pure speculation.  However, this speculation is based on the consideration that natural rubber (latex) is composed mostly of lipid with a small amountof protein as are microbial cell-surface membranes.  This ‘fact’ is what prompted the author’s speculation in an effort to comprehend how it might be possible that this compound, previously unexpected to have any utility in the treatment of Lyme disease or babesiosis might exert its action.  Indeed, the true mechanism(s) of action need to be determined by proper study by scientists who have the means to do so (and not by the treating physicians who have observed its clinical effects on patients).

Reviewer 3 suggests some emphasis be added contrasting the simplicity and relative effectiveness of a course of an inexpensive (and relatively safe) single agent for a finite duration as opposed to what had been needed previously in these patients: open-ended combinations of antimicrobials which entailed very considerable expense to the patients and to the health-care system (private insurers and government-based insurance) and much higher risk (e.g. potential complications associated with in-dwelling intravenous catheters as occurred in Case 2), inconvenience, and discomfort for patients (e.g. extended intra-muscular injection of ceftriaxone in Cases 2 and 3).  The manuscript will be revised to emphasize these points.

Round 2

Reviewer 2 Report

I do not see much improvement in study design or fact interpretation in this heavily flawed paper. The paper cannot answer the questions it poses as it was not designed to do so. Nor is it helpful to guide others.This is not a question of opinion but but of study design or to make a long story short: the difference between science and story telling.

This manuscript is a resubmission of an earlier submission. The following is a list of the peer review reports and author responses from that submission.